

RAINFALL AND ROCKFALLS IN THE CANARY ISLANDS: ASSESSING A SEASONAL
LINK
Massimo Melillo[a], Stefano Luigi Gariano[a], Silvia Peruccacci[a], Roberto Sarro[b], Rosa Marìa Mateos[c],
Maria Teresa Brunetti[a]
[a] CNR IRPI, via Madonna Alta 126, 06128, Perugia, Italia
massimo.melillo@irpi.cnr.it, stefano.luigi.gariano@irpi.cnr.it, silvia.peruccacci@irpi.cnr.it
[b] IGME, c/ Alenza, 1, 28003, Madrid, España
r.sarro@igme.es
[c] IGME, Urb. Alcázar del Genil, 4. Edificio Zulema, bajos, 18006, Granada, España
rm.mateos@igme.es
Correspondence to: Maria Teresa Brunetti maria.teresa.brunetti@irpi.cnr.it





**Abstract**
Rockfalls are frequent and harmful phenomena occurring in mountain ranges, coastal cliffs and slope
cuts. Albeit several natural processes concur in their formation and triggering, rainfall is one of the
most common causes. The prediction of rock failures is of social significance for civil protection
purposes and can rely on the statistical analysis of past rainfall conditions that caused the failures.
The paper describes the analysis of information on rainfall-induced rockfalls in Gran Canaria and
Tenerife, Canary Islands (Spain). An analysis of the monthly rainfall versus the monthly distribution
of rockfalls reveals that they are correlated for most of the year, except in summer, when other triggers
act to induce collapses. National and regional catalogues with hourly and daily rainfall measurements
are used to reconstruct the cumulated amount ($E$) and the duration ($D$) of the rainfall responsible for
the rock failures. Adopting a consolidated statistical approach, new $ED$ rainfall thresholds for possible
rockfall occurrence and the associated uncertainties are calculated for the two test sites. As far as is
known, this is the first attempt to predict this type of failure using the threshold approach. Using the
rainfall information, a map of the mean annual rainfall is obtained for Gran Canaria and Tenerife, and
it is used to assess the differences between the thresholds. The results of is study are expected to
improve the ability to forecast rockfalls in the Canary Islands, in view of implementing an early
warning system to mitigate the rockfall hazard and reduce the associated risk.

*Keywords*: Rockfall, rainfall threshold, Canary Islands.



## 1 Introduction

Rockfalls are instability processes affecting mountainous regions, coastal cliffs and slope cuts. Being very rapid, they are extremely dangerous and life-threatening, especially when they occur in populated areas, along roads and railways. The most frequent triggering factors of rockfalls are rainfall, cycling thermal stress, and seismic activity (Wieckzorek and Jaeger, 1996; Keefer, 2002; Mateos, 2016; Ansari et al., 2015; Collins and Stock, 2016; Sarro et al., 2018; Saroglou, 2019). At regional and global scales, empirical approaches to forecast the occurrence of rockfalls may contribute reducing risk. Generally, for rainfall-induced slope failures the forecast can rely upon the definition of rainfall thresholds, i.e. the rainfall conditions that when reached or exceeded are likely to trigger the failure. Rainfall thresholds are calculated through the statistical analysis of historical rainfall conditions that have resulted in landslides (e.g., Guzzetti et al., 2007, 2008; Cepeda et al. 2010; Sengupta et al., 2010; Ruiz-Villanueva et al. 2011; Berti et al., 2012; Staley et al., 2013; Zêzere et al., 2015; Palenzuela et al., 2016; Rosi et al., 2016; Peruccacci et al., 2017; Segoni et al., 2018; Valenzuela et al., 2018, 2019). The definition of reliable empirical rainfall thresholds relies on the use of objective procedures for (i) the reconstruction of the rainfall events responsible for the failures and (ii) the calculation of the thresholds. For the purpose, Melillo et al. (2018) have proposed an algorithm that reconstructs rainfall events, identifies the rainfall conditions that have resulted in slope failures, and calculates probabilistic cumulated event rainfall-rainfall duration (*ED*) thresholds at different non-exceeding probabilities and their associated uncertainties (Peruccacci et al., 2012). The obtained thresholds are a set of parallel power-law curves in a log-log (*D,E*) plane, which are characterized by a slope and an intercept, the last being a function of the non-exceeding probability value (Brunetti et al., 2010).

In this work, a relationship between the amount of rainfall and the occurrence of rockfalls is assessed and empirical rainfall thresholds are defined for two test sites in Gran Canaria and Tenerife, Canary Islands (Spain). The possible prediction of rainfall-induced rock failures is of fundamental importance primarily for the safety of the inhabitants and for preserving infrastructures such as roads and buildings. An increasing level of safety against this type of hazard is also important for the local economy, one third of which is based on tourism. As far as is known, this is the first attempt to predict rock failures triggered by rain using the threshold approach. Recently, in Italy it has been observed that the slope of the power-law curve is dependent on the mean annual rainfall (MAR). In particular, the higher is the MAR the steeper is the threshold (Peruccacci et al., 2017). This relationship is explained assuming that where the landscape has been shaped over long time periods by landslides





triggered by a given minimum amount of rainfall, it is likely necessary at least as much rainfall to
trigger the next landslides (Chen, 2015). For improving the discussion of the results, it has been
considered worthwhile producing a map of the MAR for the islands of Gran Canaria and Tenerife
using the available rainfall data sets.
The manuscript is organized as follows. After a description in Section 2 of the general settings of the
two test sites, Section 3 describes the rainfall and rockfall datasets, and the methods used to determine
*ED* rainfall thresholds and the map of the MAR. Section 4 illustrates in detail the relationship between
the rainfall regime and the occurrence of rock failures, and presents the rainfall thresholds for the
possible rockfall occurrence in the two test sites. Finally, in Section 5, the main findings of the work
are summarised and discussed.
## 2 Test site description
The Canary Islands (Spain) are one of the major volcanic chain in the oceans. The archipelago
consists of eight islands in the Atlantic Ocean, aligned along a W-SW to E-NE direction: El Hierro,
La Palma, La Gomera, Tenerife, Gran Canaria, Fuerteventura, La Graciosa and Lanzarote. The
geological origin of the Canary archipelago (800 km in length) is still under debate, but it has been
traditionally interpreted as a hotspot track (Fullea et al., 2015).
The steep topography and the geological complexity of the archipelago influence the activation of an
intense slope failures activity. Rockfalls are the most frequent landslide type in the Canary Islands,
causing damage on built-up areas and communication networks.
Two test sites are selected for assessing the relationship between the rainfall and the occurrence of
rockfalls. The first site (GC) is located in the north-western part of Gran Canaria island, and the
second site (TEN) is the entire Tenerife island.
### *2.1 Gran Canaria Island (GC-200 road)*
Gran Canaria is the third island in size of the Canarian archipelago. With an area of 1560 km$^2$ and a
maximum altitude of 1956 m a.s.l., the island is approximately circular in shape (Fig. 1). The origin
of Gran Canaria can be dated about 15 million years ago (Miocene) with the first submarine building
stages of the Gran Canaria Volcano. From the geological point of view, the island presents the greatest
variability of igneous rocks of the entire archipelago. Besides the distinctive lavas of the basanite
basalt to trachyte phonolite series, Gran Canaria presents also other types of magma, such as tholeiitic





basalts and rhyolites (Troll and Carracedo, 2016). Massive flank failures and erosion give place to
chaotic deposits that cover large areas.
The test site is the GC-200 road located in the north-western extreme of Gran Canaria, and specifically
between the localities of Agaete and Aldea. The road constitutes the main transportation corridor
between the two localities. With a length of 34 km, the road path is very tortuous following the
contour of the coast, a very step coastline with some of the highest cliffs in Europe. The road has
heavy traffic estimated on average at 1500 vehicles per day. The geology of the test site area is within
the domain of the basaltic shield stage, Middle Miocene in age. Along the road, an alternance of
alkaline basaltic deposits and piroclastic flows can be observed. In some parts, gravitational deposits
(mainly colluvial) also outcrop covering wide areas.
Regarding climatological conditions, Gran Canaria is located in a transitional zone between temperate
and tropical conditions. The conical morphology of Gran Canaria retains the humidity of the
predominant N-NE trade winds of the subtropical Azores anticyclone on the north side of the island.
As a result, the northern flanks are humid and vegetation is vigorous, while the south part of the island
is very dry and the conditions are very arid and desert-like. Annual rainfall ranges between 100 and
1000 mm on average, increasing with altitude. In the test site the climate is very dry, with low average
annual rainfall (< 100 mm) and high average annual temperature (~ 20°C).
*2.2  Tenerife island*
Tenerife (Fig. 1) is the largest (2057 km$^2$) and the most populated (950,000 inhabitants and 13.2
million visitors in 2019) island of the archipelago. It is home to the third largest volcano in the world
(Pico del Teide, 3718 m a.s.l.).
From a geological point of view, Tenerife was constructed via Miocene–Pliocene shields that now
form the vertices of the island. The shields were unified into a single edifice by later volcanism that
continued in central Tenerife from approximately 12 to 8 million years ago and was followed by a
period of dormancy. Rejuvenation at approximately 3.5 Ma is recorded by the central Las Cañadas
Volcano, and long residence times of magmas during this period favoured magmatic differentiation
processes to produce an episode of felsic and highly explosive felsic volcanism (Troll and Carracedo,

123   2016).

The steep orography of the island and the climate variety have resulted in a diversity of landscapes
and geographical formations. Very impressive coastal cliffs (till 500 m in height) are present in the





northern corner of Tenerife. This area is also characterized by narrow and deep ravines which
determine an intense slope activity.
The climate of Tenerife is subtropical oceanic; the minimum and maximum annual average
temperatures are about 15ºC in winter and 24ºC in summer. Tenerife offers a large variety of micro-
climate zones controlled by the altitude and the winds.

## 3   Data and methods


The availability of rainfall measurements and landslide information is fundamental to define reliable
rainfall thresholds. For the selection of the rain gauges, the data quality and the location of the rain
gauges are assessed, given that these features are crucial to characterize the spatial-temporal variation
of the precipitations. Similarly, the calculation of the MAR relies on the availability of sufficiently
long rainfall series (at least 30 years). This is difficult to achieve for a dense network of rain gauges,
where sensors may exhibit different operating time periods. The World Meteorological Organization
(WMO) guidelines on the calculation of the annual standard normal, specifically the MAR,
recommend at least 10 years to define at least provisional MAR maps (WMO, 1989). This is the case
in the test sites, where a lot of rainfall information is limited to short time periods (the average is 15.6
years), thus hampering the calculation of the MAR with a detailed space resolution.

### *3.1  Rainfall data*


In the GC test site, hourly rainfall data (purple triangles in Fig. 1) from the Spanish National
Meteorological Service (AEMET) network (in total 25 stations among which 4 are close to the study
area) are used for the calculation of rainfall thresholds. Moreover, daily rainfall data (orange triangles
in Fig.1) are provided from the *Consejo Insular de Aguas de Gran Canaria* (CIAGC) regional rain
gauge network (13 stations) and from AEMET (92 stations, among which 7 are close to the study
area). Some of the sensors of the AEMET network provide both hourly and daily rainfall, in different
time periods. Details of the rainfall series are reported in Table 1.
For the TEN test site, rainfall measurements are provided by AEMET with the contribution of
regional networks. As for the GC test site, the rainfall analysis is performed using both hourly and
daily data. The two networks in the TEN test site are composed by 34 rain gauges recording hourly
data (purple triangles in Fig. 1) and 66 rain gauges recording daily data (orange triangles in Fig. 1).



To calculate the MAR for the two test sites, yearly and monthly rainfall data provided by AEMET
and by *Sistema de Información Agroclimático y de Regadíos* (SIAR), respectively, are used (Table
1). In particular, in order to obtain homogeneous maps, data recorded in the 20-year period from
January 2000 to December 2019 in both test sites are selected. Following WMO guidelines (WMO,
1989), only stations with at least 10 years of data are included in the analysis. Overall, 72 (one every
22 km$^2$) and 67 (one every 31 km$^2$) rain gauges are used to calculate MAR in Gran Canaria and
Tenerife, respectively. The average number of sensors operating per year in the considered period is
56 (84%; one every 28 km$^2$) in Grand Canaria and 47 (65%; one every 43 km$^2$) in Tenerife. The used
rain gauges are homogeneously distributed over the test site areas.
Using the monthly and annual rainfall data recorded by the 103 rain gauges in the two islands, the
MAR for the period 2000-2019 was calculated for each station. Moreover, the coefficient of variation
of the MAR is calculated by dividing its standard deviation by the MAR. This coefficient represents
the variability of the MAR in the considered time interval. The map of the MAR and of its coefficient
of variation are calculated using the tension spline tool in ESRI ArcMAP 10.7.1.

### 3.2  Rockfall data

The information on the rockfalls was collected by the Canarian Civil Protection Authorities in the
TEN test site and by the Road Maintenance Service in the GC test site. In particular, for the GC test
site a total of 8174 rockfall events occurred from January 2010 to March 2016 was documented. A
catalogue was prepared defining accurately the location of each impact along the road using
orthophotos available for the region and technical reports. The information for each event includes
kilometre point, number of events, date, and boulder size. In GC only 535 rockfalls characterized by
medium to large size are included in the analysis for the thresholds, whereas small and very small
rockfalls ($< 10^{-3}$ m$^3$) are discarded. Analogously, a catalogue of 1898 rockfalls that impacted along
Tenerife roads from January 2010 to November 2017 was prepared. For each event, the information
includes rockfall localization, geographic accuracy, occurrence day, month, year, and time (if
available), and temporal accuracy.
The influence of the rainfall on the occurrence of rockfalls is assessed analysing the distribution of
monthly rainfall (Figs. 2a,b) and monthly number of rockfalls (Figs. 2c,d) on the two test sites. As
expected, an increase of the rainfall in the autumn-winter period, between October and March, is
observed in both islands, with a maximum in November.



The monthly distribution of rockfalls in Gran Canaria (Fig. 2c) is coherent with the rainfall values in
the period January-April, with a maximum in February (~ 130; Fig. 2a). For the remaining dry (May
to September) and wet (October to December) months the number of rock failures decreases and
becomes almost flat (below 50). This behaviour suggests the presence of triggering mechanisms other
than the rainfall. For the TEN test site, the number of rockfalls per month (Fig. 2b) is similar to the
rainfall distribution, confirming anyway the presence of one or more additional triggers as evidenced
by the abundance of failures between May and September (Fig. 2d) when the rainfall is irrelevant.
*3.3 Empirical rainfall thresholds*
Empirical *ED* thresholds are represented by the following power law curve:
$$E = (\alpha \pm \Delta\alpha) \times D^{(\gamma \pm \Delta\gamma)} \qquad (1)$$
where *E* is the cumulated event rainfall (in mm), *D* is the duration of the rainfall event (in hours or in
days), $\alpha$ and $\gamma$ are the intercept and the slope of the curve, respectively, and $\Delta\alpha$ and $\Delta\gamma$ are the
uncertainties associated with them. Thresholds at different non-exceedance probabilities are
calculated adopting the frequentist approach and the bootstrap nonparametric statistical technique
(Brunetti et al., 2010; Peruccacci et al., 2012), using 5000 randomly selected synthetic series of *DE*
pairs. A threshold at 5% non-exceedance probability should leave 5% of the empirical *DE* pairs below
itself. The parameter uncertainties depend mostly on the number and the distribution of the rainfall
conditions. The minimum number of *DE* pairs needed for having stable mean values of the parameters
$\alpha$ and $\gamma$ (i.e. reliable thresholds) depends on the distribution and dispersion of the empirical data points
in the *DE* domain.
*3.4 The CTRL-T algorithm for threshold calculation*
The quantitative identification of the rainfall responsible for slope failures and the definition of
reliable thresholds are fundamental steps towards a well-founded event prediction (Peruccacci et al.,
2017; Melillo et al., 2018). The use of standardized procedures for the reconstruction of the rainfall
conditions able to trigger past failures and for the definition of thresholds is necessary for enhancing
the objectivity and reproducibility of the curves. The tool named CTRL-T (Calculation of Thresholds
for Rainfall-induced Landslides - Tool) proposed by Melillo et al. (2018) is exploited to calculate *ED*
thresholds for the two test sites. CTRL-T reconstructs rainfall events starting from continuous rainfall
series. For each rockfall, the algorithm: 1) identifies automatically the representative rain gauge; 2)
identifies multiple (*D,E*) rainfall conditions responsible for the failure; 3) selects among them the




maximum probability rainfall conditions (MPRCs). Then, analysing the distribution of the MPRCs it
calculates probabilistic rainfall thresholds at different non-exceeding probabilities and their
associated uncertainties. In order to avoid using wrong temporal information (i.e., incorrect dates for
the occurrence of rockfalls) in the definition of the thresholds, the rainfall conditions having a delay
longer than 48 hours between the rainfall ending time and the rockfall occurrence are discarded.
Using CTRL-T, 82 rockfalls occurred between 2012 and 2016 in GC test site and 626 rockfalls
occurred between 2010 and 2016 in the TEN test site are selected (light green dots in Fig. 2). The
remaining records are discarded due to the: 1) absence of rainfall data in the period including the
collapse occurrence time; 2) absence of rain gauges within a buffer of 15 km radius centred on the
rockfall; 3) lack of an evident correlation with the rainfall. The definition of rainfall thresholds relies
only upon rainfall conditions that triggered the first failure in each event. As a consequence, numerous
rockfalls (106, 39% in GC and 271, 30% in TEN) which occurred at the same date and in the same
location, and which are associated with the same rainfall event are discarded. In GC among the
remaining rockfalls 53 are analysed with daily and 29 with hourly rainfall data, respectively. The low
number of rock failures associated to hourly-based rain gauges is to be ascribed to the low density of
the sensors in the area. In TEN 245 rockfalls are reconstructed with hourly data and 381 with daily
rainfall data. Note that for 83 failures it was possible reconstructing the rainfall conditions using
sensors from both the two rain gauge networks. As a consequence, the reconstructed ($D$,$E$) rainfall
conditions have different temporal resolutions and are used to define both hourly-based and daily-
based rainfall thresholds.
**4    Results**
A correlation between the rainfall and the observed failures is confirmed by the comparison between
the monthly rainfall and the corresponding number of rockfalls both in GC and in TEN (Fig. 3).
Figures 3a,b,c show the boxplots of cumulated monthly rainfall based on the data recorded in rain
gauges used to reconstruct the rainfall responsible for rockfalls for GC and TEN test sites. Inspection
of these figures reveals that the rainfall pattern in the two test sites is typically Mediterranean, with a
maximum in winter (but also in October and November) and a minimum in summer, with practically
no rain in the warmest months. Analysing data from seven daily-based rain gauges in GC (GC-d), it
turns out that the rainiest months are February and November with an average rainfall of 52.2 mm
and 55.7 mm, a highest rainfall of 98.6 mm and 133.9 mm, and a median rainfall of 42.3 mm and
39.8 mm, respectively (Fig. 3a). A similar trend is found for Tenerife using both daily and hourly
data. Data from 40 daily-based rain gauges in TEN (TEN-d) are analysed finding an average rainfall





of 64.6 mm and 86.4 mm, a highest rainfall of 183.5 mm and 183.6 mm, and a median rainfall of 56.1
mm and 93.2 mm for February and November, respectively (Fig. 3b). Data from 21 hourly-based rain
gauges in TEN (TEN-h) are analysed finding an average rainfall of 88.8 mm and 82.0 mm, a highest
rainfall of 169.8 mm and 190.8 mm, and a median rainfall of 97.5 mm and 66.4 mm for February and
November, respectively (Fig. 3c).
Figures 3d,e,f portray the monthly number of rockfalls associated with rainfall events for GC-d, TEN-
d and TEN-h. The GC catalogue lists 53 collapses occurred in the period from November 2012 to
October 2016, with the majority of the failures in 2015 (22). The month with the largest number of
rockfalls (14) is February, followed by January (8) and November (7). The least number of failures
is reported in September (1) and no rainfall-induced rockfalls are reported in May and July (Fig. 3d).
The 245 rock failures in the TEN-d catalogue cover the period from September 2010 to February
2016, with the majority of records in 2014 (66). The month with the largest number of rockfalls (80)
is November, followed by October (37) and December (36). The least number of failures is reported
in May (1) and no rainfall-induced rockfalls are reported in June and July (Fig. 3e). The TEN-h
catalogue lists 381 rockfalls occurred in the period from September 2010 to November 2016, with
the majority of the failures in 2014 (90). The month with the largest number of rockfalls (115) is
November, followed by December (72) and October (64). The least number of failures is reported in
May (1) and no collapses are reported in July (Fig. 3f).
The rainfall that triggered the rockfalls is classified according to the method proposed by Alpert et al.
(2002), based on six daily rainfall ($E_d$) categories from "light" to "torrential" over the Mediterranean
(Table 2). Using the procedure adopted by Melillo et al. (2016), each rainfall condition responsible
for rock failures (MPRC) is attributed to a specific category. In particular, for events lasting less than
24 hours, a category based on the total cumulated rainfall of the event is assigned. For events lasting
more than 24 hours, the maximum value of the cumulated rainfall in 24 hours in a moving window is
used. In GC, over 40% of the MPRCs responsible for the collapses are classified as moderate-high
(MH); in TEN, approximately 30% as high (H) and high-torrential (HT). No MPRCs are found in the
lowest Alpert's category (light, Table 2). Figures 3g,h,i show the cumulated percentage of rainfall
events per month grouped according to Alpert's classification. In GC-d, in February (Fig. 3g) 6
rockfalls (43%) are triggered by a rainfall classified as H, 3 (21%) as torrential (H), 3 (22%) as MH,
and 1 as light-moderate (LM) and HT each (14%). In TEN-d, in November, 29 rockfalls (36%) are
triggered by a rainfall classified as HT, 26 (33%) as MH, 22 (28%) as H, 2 (2%) as LM, and 1 (1%)
as T (Fig. 3h). In TEN-h, in November, 5 (4%), 26 (23%), 26 (23%), 31 (27%) and 27 (23%) rockfall
are triggered by a rainfall classified as LM, MH, H, HT and T, respectively (Fig. 3i).



Using the catalogues of rainfall events with rockfalls described above and the CTRL-T tool, *ED*
thresholds, and their associated uncertainties are calculated for GC and TEN test sites. Table 3 lists
the number of MPRC used to define the thresholds, the equations of the power law curves, and the
range of validity for the thresholds, expressed in hours or days. Note that *D* must be expressed in days
in the equations for the thresholds calculated with daily data, and in hours in the equations for the
thresholds calculated with hourly data (Gariano et al., 2020).
Figure 4a shows, in logarithmic coordinates, the distribution of the (*D,E*) rainfall conditions,
reconstructed with daily data, that have caused rockfalls in GC (53 blue dots) and in TEN (245 green
dots). In particular, the 53 daily rainfall conditions responsible for the rockfalls in GC have durations
in the range $1 \leq D \leq 11$ days (with an average value of 2 days) and cumulated rainfall in the range
$16.5 \leq E \leq 219.9$ mm (average value 51.6 mm). All the conditions were recorded in rain gauges
located at a maximum distance of 5.7 km from the failures, with a mean value of 2.8 km. The 245
daily-based rainfall conditions associated with the collapses in TEN have durations ranging from one
to 15 days, with a mean value of two days. The cumulated rainfall ranges from 15.4 to 235.0 mm,
with an average of 71.5 mm. The average distance between the rockfalls and their representative rain
gauges is 2.2 km, with a maximum distance of 5 km. Figure 4a portrays also the 5% *ED* thresholds
for GC ($T_{5,GC-d}$, blue curve) and TEN ($T_{5,TEN-d}$, green curve). The shaded areas around the threshold
lines show the uncertainty regions associated to the thresholds (Table 3). Figure 5b portrays the same
$T_{5,GC-d}$ and $T_{5,TEN-d}$, in linear coordinates, in the range $1 \leq D \leq 7$ days.
Figure 4c shows, in logarithmic coordinates, the distribution of the (*D,E*) rainfall conditions,
reconstructed with hourly data, that have triggered rock failures in TEN (381 purple dots). The hourly
rainfall conditions associated to rockfalls have durations ranging from 2 to 712 hours and mean value
of 111 hours. The cumulated rainfall ranges from 10.6 to 433.9 mm, with an average of 105.6 mm.
The average distance between the rockfalls and the representative rain gauges is 6.7 km, with a
maximum distance of 14.9 km. In the log-log plot the purple curve is the 5% threshold for TEN
($T_{5,TEN-h}$) obtained with hourly data. Figure 5d portrays the same $T_{5,TEN-h}$, in linear coordinates, in the
range $1 \leq D \leq 120$ hours. The uncertainty associated with the threshold (purple shaded area in Figs.
4c,d) is also shown.
The difference between the $T_{5,GC-d}$ and $T_{5,TEN-d}$ thresholds can be ascribed to the different MAR in the
two test sites. Figure 5 portrays the maps of the MAR and of its coefficient of variation, which is the
percentual variability (standard deviation) of the MAR in the considered time interval. The
geographical distribution of the MAR values exhibits the highest values in the northern parts of both





islands, where it overcomes 800 mm (Fig. 5a). On the contrary, the highest values of the coefficient
of variation (i.e. an index of the MAR variability) are localized in the southern part of the islands,
where the rain gauge density is lower (Fig. 5b).
**5    Discussion and conclusions**
In Canary Island rainfall is the most important triggering factor for rockfalls (Fig. 2). Nevertheless,
there are other factors that predispose directly or indirectly the trigger of the failure (Temiño et al.,
2013a). Factors that greatly accentuate this hazard in the two test sites are wind, geomorphological
characteristics (e.g., slope, aspect), type of soil and seismic activity. Regarding the wind many
collapses are caused by strong gusts of wind that affect the northern side of Tenerife Island and the
road GC-200 from Agaete to Aldea in Gran Canaria. (Temiño et al., 2013b). Regarding the
geomorphology, the existence of many sections of road running through the old basaltic massifs with
significant sub-vertical jointing, makes the area very susceptible to rock failures. In addition, the
action of the trade winds on the higher altitude areas, produces an increase in the relative humidity,
as large masses of water vapor are retained by steep slopes resulting in an intense weathering (and
weakening) of the rock masses. Finally, the large flank instability of the two test sites (especially in
the northwest sector of the Gran Canaria island) could be related to structural control and to seismic
activity connected to the dynamic geologic condition that characterizes them. (Masson et al., 2002;
Temiño et al. 2013b; Urgeles et al., 2001).
By selecting the subset of rockfalls triggered by rainfall it can be observed that their monthly
frequency is linked to the monthly distribution of the rainfall measured in nearby rain gauges (Figs.
3a-f). For GC-d (Figs. 3a,d) the correlation is apparently weaker in fall than in winter, but this could
be ascribed to a statistical fluctuation and should be confirmed by increasing the number of events.
Conversely, for TEN-d (Figs. 3b,e) the monthly number of rock failures well reflects the monthly
rainfall amount, suggesting that rainfall is the only triggering cause. Hourly rainfall data in TEN-h
(Figs. 3c,f) confirm partially this outcome, since even with a median lower amount of rainfall, a
higher number of rock failures is expected to occur from October to December than in February.
The number of rockfalls for which it has been possible to reconstruct the rainfall conditions (MPRCs)
using daily and hourly data in the TEN test site (Figs. 3e,f) is different. This is mostly due to the worst
temporal resolution of the TEN-d dataset.



In the two test sites, the majority of the rainfall responsible for rockfalls belongs to the Alpert's MH
category (Figs. 3g,i). In TEN-h, 31 events belong to the most severe category T, whereas in TEN-d
only one event is found in the T category. This result could be ascribed to the time step of the moving
window used to assign the Alpert's category. For a rainfall event lasting more than one day, the
Alpert's category varies depending on the data temporal resolution, since the time step is one hour or
one day for the hourly and daily data, respectively. In TEN-d, the total amount of rainfall responsible
for the failure is shared in two or more consecutive days, causing a lowering of the Alpert's category,
as confirmed by the paucity of T events in TEN-d.
Figure 4 shows that $T_{5,TEN-d}$ is higher and steeper than $T_{5,GC-d}$. This means that, at increasing values
of $D$, a smaller amount of rainfall ($E$) is necessary to trigger the collapses in GC than in TEN.
Comparing Figures 1 and 5, the recorded rockfalls in the TEN test site are localized in areas including
several classes of MAR (ranging from 100 to 800 mm), while in GC test site they fall in the area
characterized by the lowest class of MAR ($\leq$ 100 mm). The different ranges of MAR values in the
two test sites are able to explain the observed differences in the two daily $ED$ thresholds (Fig. 4a).
This finding confirms that where the MAR is higher, the minimum rainfall conditions able to trigger
a failure, specifically a rockfall, are also higher.
Moreover, the threshold defined for the TEN test site has an uncertainty smaller than the threshold
for GC test site. Peruccacci et al. (2012) observed that the parameter uncertainty reduces as the
number of MPRC used to calculate the threshold increases. In particular, as derived from Table 3, the
relative uncertainty of the intercept, $\Delta\alpha/\alpha$ is 9.8% for $T_{5,GC-d}$ and 4.9% for $T_{5,TEN-d}$. Regarding the
slope of the curves, $\Delta\gamma/\gamma$ is 16.1% for $T_{5,GC-d}$ and 6.7% for $T_{5,TEN-d}$. Given the lower uncertainty range
and relative uncertainties of both parameters, $T_{5,TEN-d}$ has a reliability higher than that of $T_{5,GC-d}$. The
same analysis for the $T_{5,TEN-h}$ threshold gives $\Delta\alpha/\alpha = 9.3\%$ and $\Delta\gamma/\gamma = 4.2$. Thresholds with an hourly
temporal resolution and having relative uncertainties of the parameters $\alpha$ and $\gamma$ lower than 10% could
be implemented in an operative system for the prediction of rainfall-induced failures (Peruccacci et
al., 2012; 2017). The thresholds for different non-exceedance probabilities obtained for TEN test site
using hourly rainfall data are suited for the design of probabilistic schemes for the operative prediction
of rainfall-induced rockfalls. An improvement in the number of rain gauges providing hourly
measurements, as well as in the number of recorded rock failures, would be necessary in GC test site
in order to reduce the uncertainty of the threshold.
Currently, neither prototype nor operative early warning systems for rainfall-induced failures are
present in the Canary Islands (Guzzetti et al. 2020). The findings of this work can contribute to the



understanding of the rainfall conditions that can trigger rainfall-induced rockfalls in Tenerife and in
the western part of Gran Canaria, and their relationship with mean annual rainfall regime. These
findings have scientific and social implications given that in both test sites also spring and autumn
are characterized by a moderately occurrence of rock failures, with relevant impacts on the
population, tourism activities, and local economy. As long as a sufficient amount of empirical data
will be available in both test sites (and also in other islands of the archipelago), the method adopted
in this work for the definition of reliable rainfall thresholds can be replicated, and the results can be
implemented in a prototype early warning system.
## 6  Acknowledgements
Research conducted within the framework of the U-Geohaz project (Geohazard Impact Assessment
for Urban Areas) funded by the European Commission, Directorate-General Humanitarian Aid and
Civil Protection (ECHO), under the call UCPM-2017-PP-AG. This work was also funded by the
Salvador de Madariaga Mobility Program from the Spanish Ministry of Science, Project:
PRX18/00020.

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


---

Rainfall and rockfalls in the Canary Islands: assessing a seasonal link

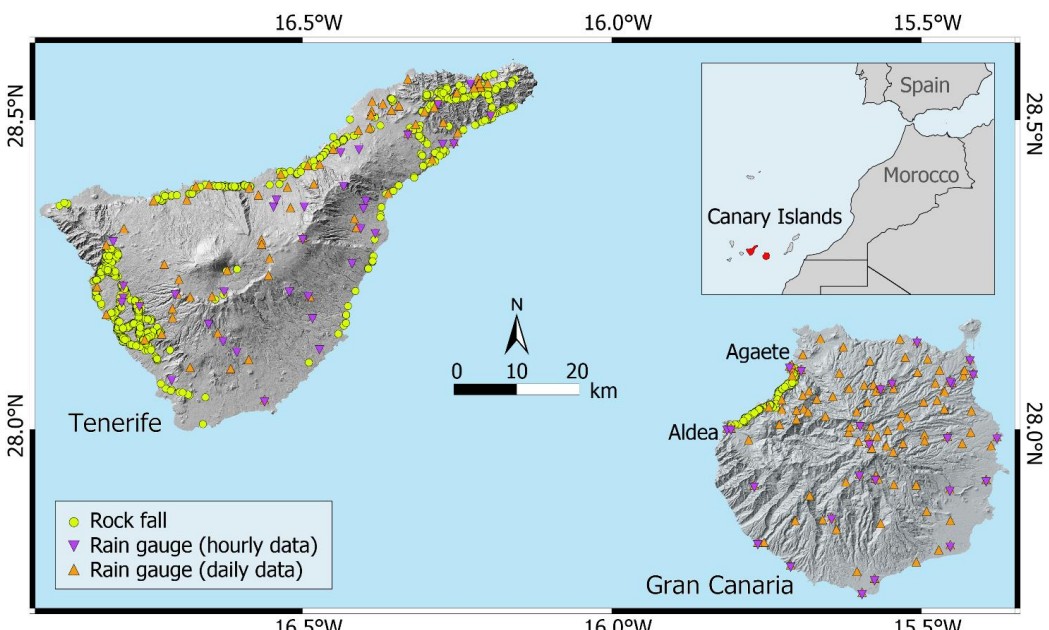


Figure 1. GC and TEN test sites. Location of the rain gauges providing hourly (purple triangles) and daily
(orange triangles) rainfall measurements, and of rockfalls used for threshold calculations (light green dots).
Hillshade derived from MDT05 2009 CC-BY 4.0 scne.es.
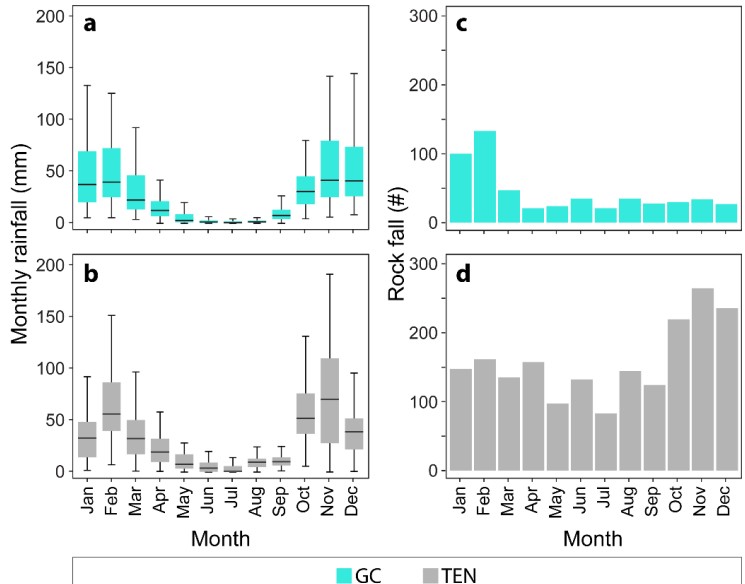


Figure 2. Comparison between monthly rainfall and rockfall occurrence. (a, b) Annual variation of monthly

rainfall measures in GC (cyan) and TEN (grey). The whiskers show 1.5 times the interquartile range. (c, d)

Number of rockfalls per month in the two test sites.
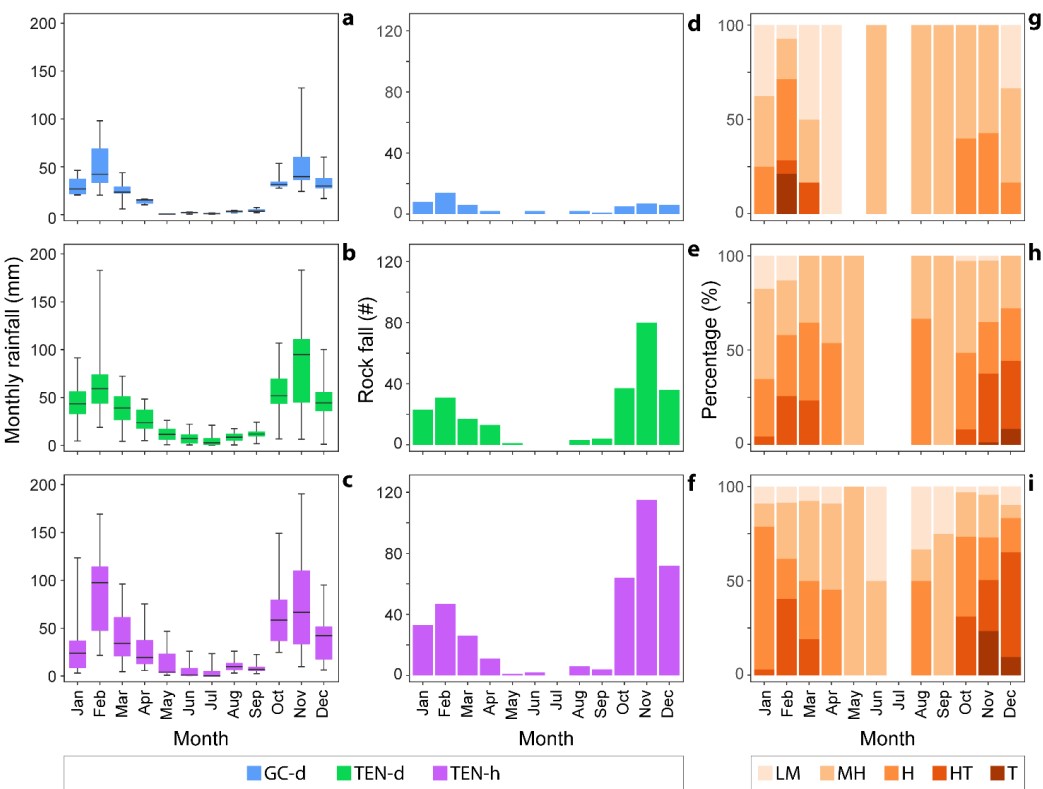

Figure 3. Comparison between monthly rainfall and rainfall-induced rockfalls and Alpert classification. (a, b, c) Annual variation of monthly rainfall measures in the test sites. Legend: GC-d, daily rainfall data in GC test site; TEN-d, daily rainfall data in the TEN test site; TEN-h, hourly rainfall data in TEN test site. (d, e, f) Number of rainfall-induced rockfalls per month. (g, h, i) Cumulated percentage of rainfall events per month classified according to Alpert et al. (2002). Legend: LM, light-moderate ($4 < E_d \leq 16$ mm); MH, moderate-heavy ($16 < E_d \leq 32$ mm); H, heavy ($32 < E_d \leq 64$ mm); HT, heavy-torrential ($64 < E_d \leq 128$ mm); T, torrential ($E_d > 128$ mm).



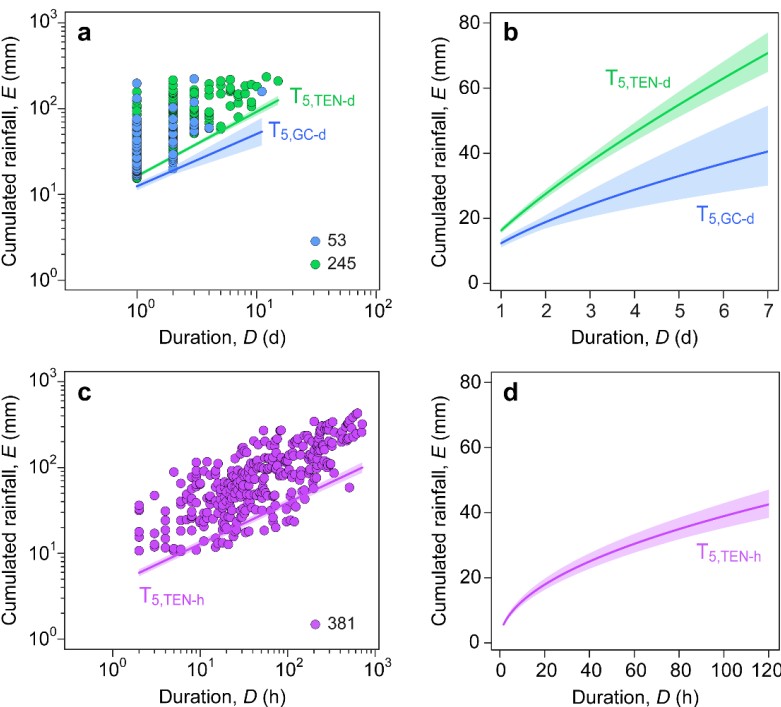

512

Figure 4. Rainfall thresholds for the possible rockfall occurrence in the two test sites. (a) Rainfall duration $D$
(x-axis, in days) and cumulated event rainfall $E$ (y-axis, in mm) conditions that have produced rockfalls in GC
(53 blue dots) and TEN (245 green dots) test sites, respectively. Green and blue curves are the 5% power law
thresholds ($T_{5,TEN-d}$, $T_{5,GC-d}$). (b) 5% daily $ED$ thresholds for GC and TEN in linear coordinates, in the range of
durations $1 \leq D \leq 7$ days. (c) Rainfall duration $D$ (x-axis, in hours) and cumulated event rainfall $E$ (y-axis, in
mm) conditions that have produced rockfalls in TEN (381 purple dots) test site. Purple curve is the 5% power
law threshold ($T_{5,TEN-h}$). (d) 5% hourly $ED$ thresholds for GC and TEN in linear coordinates, in the range of
durations $1 \leq D \leq 120$ hours.




Rainfall and rockfalls in the Canary Islands: assessing a seasonal link

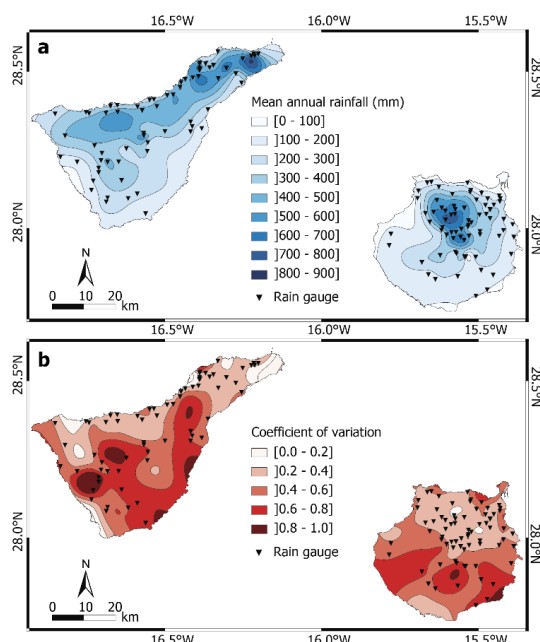


Figure 5. Maps of (a) mean annual rainfall and (b) of its coefficient of variation. The rain gauges used for these
analysis (cf. Table 1) are also shown.





Table 1. Summary of the three available rain gauge networks (CIAGC, AEMET, SIAR) in the two test sites
(GC and TEN) i.e., network name, network operating time period, temporal resolution, test site, number of
used rain gauges, their average operating time, and the use of data.

| Network | Period | Temporal resolution | Test site | Rain gauges (#) | Average operating time (year) | Data application |
|---|---|---|---|---|---|---|
| CIAGC | Jan 2010 - Dec 2017 | daily | GC | 13 | 8.0 | Thresholds |
| AEMET | Jan 1951 - May 2019 | daily | GC | 92 | 41.8 | |
| | Oct 1997 - May 2019 | hourly | GC | 25 | 16.5 | |
| | Jan 2010 - Mar 2018 | hourly | TEN | 34 | 5.5 | |
| | Jan 2010 - May 2018 | daily | TEN | 66 | 8.2 | |
| | Jan 2000 - Dec 2019 | yearly | GC | 67 | 15.2 | MAR |
| | | | TEN | 58 | 13.8 | |
| SIAR | Jan 1999 - Dec 2019 | monthly | GC | 5 | 18.2 | |
| | | | TEN | 9 | 15.1 | |




Table 2. Summary of the number (#) and percentage (%) of MPRC in the categories proposed by Alpert et al.
(2002), in the two test sites.

| Category | $E_d$ (mm) | GC-d | | TEN-d | | TEN-h | |
|---|---|---|---|---|---|---|---|
| | | # | % | # | % | # | % |
| Light (L) | $E_d \leq 4$ | 0 | 0 | 0 | 0 | 0 | 0 |
| Light-moderate (LM) | $4 < E_d \leq 16$ | 11 | 20.7 | 11 | 4.5 | 28 | 7.3 |
| Moderate-heavy (MH) | $16 < E_d \leq 32$ | 23 | 43.4 | 92 | 37.5 | 86 | 22.6 |
| Heavy (H) | $32 < E_d \leq 64$ | 14 | 26.4 | 80 | 32.7 | 117 | 30.7 |
| Heavy-torrential (HT) | $64 < E_d \leq 128$ | 2 | 3.8 | 58 | 23.7 | 116 | 30.5 |
| Torrential (T) | $E_d > 128$ | 3 | 5.7 | 4 | 1.6 | 34 | 8.9 |





Table 3. *ED* rainfall thresholds at different non-exceedance probabilities (1%, 5%, 10%, 20%, 35% and 50%)
for the GC and TEN test sites. The number of MPRC and the duration range of each threshold are also reported.

| Threshold name | Number of MPRC | Threshold equation | Duration range |
|---|---|---|---|
| $T_{1,GC\text{-}d}$ | | $E = (8.3\pm1.0)\times D^{(0.62\pm0.10)}$ | |
| $T_{5,GC\text{-}d}$ | | $E = (12.3\pm1.2)\times D^{(0.62\pm0.10)}$ | |
| $T_{10,GC\text{-}d}$ | 53 | $E = (15.1\pm1.4)\times D^{(0.62\pm0.10)}$ | 1-11 days |
| $T_{20,GC\text{-}d}$ | | $E = (19.5\pm1.8)\times D^{(0.62\pm0.10)}$ | |
| $T_{35,GC\text{-}d}$ | | $E = (25.5\pm2.5)\times D^{(0.62\pm0.10)}$ | |
| $T_{50,GC\text{-}d}$ | | $E = (31.9\pm3.6)\times D^{(0.62\pm0.10)}$ | |
| $T_{1,TEN\text{-}d}$ | | $E = (11.6\pm0.6)\times D^{(0.75\pm0.05)}$ | |
| $T_{5,TEN\text{-}d}$ | | $E = (16.3\pm0.8)\times D^{(0.75\pm0.05)}$ | |
| $T_{10,TEN\text{-}d}$ | 245 | $E = (19.6\pm0.8)\times D^{(0.75\pm0.05)}$ | 1-15 days |
| $T_{20,TEN\text{-}d}$ | | $E = (24.4\pm1.0)\times D^{(0.75\pm0.05)}$ | |
| $T_{35,TEN\text{-}d}$ | | $E = (30.6\pm1.4)\times D^{(0.75\pm0.05)}$ | |
| $T_{50,TEN\text{-}d}$ | | $E = (37.1\pm1.8)\times D^{(0.75\pm0.05)}$ | |
| $T_{1,TEN\text{-}h}$ | | $E = (2.8\pm0.3)\times D^{(0.48\pm0.02)}$ | |
| $T_{5,TEN\text{-}h}$ | | $E = (4.3\pm0.4)\times D^{(0.48\pm0.02)}$ | |
| $T_{10,TEN\text{-}h}$ | 381 | $E = (5.3\pm0.5)\times D^{(0.48\pm0.02)}$ | 2-712 hours |
| $T_{20,TEN\text{-}h}$ | | $E = (6.9\pm0.6)\times D^{(0.48\pm0.02)}$ | |
| $T_{35,TEN\text{-}h}$ | | $E = (9.1\pm0.7)\times D^{(0.48\pm0.02)}$ | |
| $T_{50,TEN\text{-}h}$ | | $E = (11.4\pm1.0)\times D^{(0.48\pm0.02)}$ | |
