# Peer review of "RAINFALL AND ROCKFALLS IN THE CANARY ISLANDS: ASSESSING A SEASONAL"

_Natural Hazards and Earth System Sciences, 2020_

## Referee Comment (RC1) · Anonymous Referee #1 · 20 May 2020

Nice study. Nicely and well written. Everything's fine.

Very small comments:

- The first two lines in both abstract and introduction theoretically could be left out. These sentences are found in almost every rockfall publication.

- Line 88: you could add a reference to Fig. 1 here as well.

- Section 3.2: how were the occurance times of the rockfalls been detected? Was it the date of the road inspection? How often took the inspection place? etc.

- Section 5: is there a reasonable approach to see rainfall intensity changing due to climate change and therefore to use this study to predict rockfall activity?

[Figure]

- References: You could add Contino et al, NHESS 17(12), 2017.

-

---

## Referee Comment (RC2) · Anonymous Referee #2 · 28 May 2020

This manuscript applies the ED rainfall threshold on rockfall and discusses the link between the change of seasonal rainfall amount and the occurrence of rockfall through great amounts of rockfall cases and rainfall data. The contents, figures, tables are all well-organized. The results also provide a new insight for the early warning of rockfall. Here are some suggestions for the authors that may take into account.

1. Since there are lots of rockfall records in GC and TEN, is it possible to discuss the influence of geology and geomorphology on rainfall threshold? Besides, the relationship between the volume of rockfall and the cumulated event rainfall may also be discussed.

2. [ Section 2.2, Line 128~130 ] Comparing to the last paragraph in section 2.1, the information of average annual rainfall in TEN is missed.

3. [ Figure 4 & Line 227 ] There are 29 cases with hourly rainfall data in GC, but they are not analyzed in Figure 4 c and d. Too less to be analyzed?

4. It is suggested to add some representative pictures of these rockfalls in order to make readers understand the environment of study area more.

Minor issues:

[ Line 219∼220 ] "light green dots in Fig. 2" These are no green dots in Figure 2. Is it Figure 4?

[Figure 2 and Figure 3] The Y-axis of Fig. 2 c, d and Fig. 3 d, e, f should be "Rockfall" instead of "Rock fall."

---

## Author Comment (AC1) · 1 Jun 2020

**Reviewer #1**

Nice study. Nicely and well written. Everything's fine.

R: we thank Reviewer #1 for his/her kind comments.

**Very small comments:**

- The first two lines in both abstract and introduction theoretically could be left out. These sentences are found in almost every rockfall publication.

R: We thank for this comment. Nevertheless, we would prefer to leave these generic statements to introduce the topic and the potential hazard of rockfalls in the wider framework of rainfall-induced mass movements.

- Line 88: you could add a reference to Fig. 1 here as well.

R: Done.

- Section 3.2: how were the occurance times of the rockfalls been detected? Was it the date of the road inspection? How often took the inspection place? etc.

R: We thank the reviewer for the question that allowed us to clarify the item. We added now the following text in Section 3.2: "*After a rockfall occurrence, a quick action is required by local authorities to remove boulders from road, and to repair the damage. The average response time for these emergencies is down to one day, and therefore the date of the rockfall occurrence is the same of the road inspection.*"

Apart from scheduled inspections, a survey is carried out whenever a rockfall occurrence is reported.

- Section 5: is there a reasonable approach to see rainfall intensity changing due to climate change and therefore to use this study to predict rockfall activity?

R: This is an interesting topic, indeed. Following the indications of the World Meteorological Organization, which reads: "a 30-year period is long enough to filter out any inter annual variation or anomalies, but also short enough to be able to show longer climatic trends", we believe that the number of years available to us to assess a reliable climatic trend is inadequate.

- References: You could add Contino et al, NHESS 17(12), 2017.

R: Done.

---

## Author Comment (AC2) · 5 Jun 2020

**Reviewer #2**

This manuscript applies the ED rainfall threshold on rockfall and discusses the link between the change of seasonal rainfall amount and the occurrence of rockfall through great amounts of rockfall cases and rainfall data. The contents, figures, tables are all well-organized. The results also provide a new insight for the early warning of rockfall. Here are some suggestions for the authors that may take into account.

R: We thank the Reviewer #2 for his/her kind comments.

1. Since there are lots of rockfall records in GC and TEN, is it possible to discuss the influence of geology and geomorphology on rainfall threshold? Besides, the relationship between the volume of rockfall and the cumulated event rainfall may also be discussed.

R: We thank the Reviewer #2 for the interesting suggestion, but from the geological point of view, all Canary Islands are located inside the Africa Plate, in the transitional zone between the Continental Crust and the Oceanic Crust, and are characterized by the volcanic origin that is still active. As a consequence, there aren't significant differences in the geology. Moreover, rockfalls in our catalogue occurred on the road network of GC and TEN, and therefore the dependence of the thresholds on the geology and geomorphology is likely not relevant here.

Regarding the second question, besides a qualitative classification in small, medium, and large boulders provided by the Civil Protection Authorities, the volume associated to each rockfall is unfortunately unavailable. We agree with the Reviewer that this relationship would have been very useful in defining the magnitude of the event as a function of the rainfall.

2. [ Section 2.2, Line 128~130 ] Comparing to the last paragraph in section 2.1, the information of average annual rainfall in TEN is missed.

R: We now added the required information for TEN, and we corrected the upper average annual rainfall.

3. [ Figure 4 & Line 227 ] There are 29 cases with hourly rainfall data in GC, but they are not analyzed in Figure 4 c and d. Too less to be analyzed?

R: The Reviewer #2 is right. The low number of cases precludes the calculation of rainfall thresholds. We now added this statement in the text.

4. It is suggested to add some representative pictures of these rockfalls in order to make readers understand the environment of study area more.

R: We thank Reviewer #2 for this suggestion. We now added two pictures of rockfalls on GC-200 road and on a road in Tenerife (shown below).

[Figure]

[Figure]

**Minor issues:**

[ Line 219~220 ] "light green dots in Fig. 2" These are no green dots in Figure 2. Is it Figure 4?

R: Light green dots are in Fig.1 We corrected the citation.

[Figure 2 and Figure 3] The Y-axis of Fig. 2 c, d and Fig. 3 d, e, f should be "Rockfall" instead of "Rock fall."

R: We thank Reviewer, and we noticed that the same error was also in Fig. 1. We corrected it.